# Clonal diversification and histogenesis of malignant germ cell tumours

Thomas R. W. Oliver [1,2,12], Lia Chappell [1,12], Rashesh Sanghvi [1], Lauren Deighton[1], Naser Ansari-Pour [3,4], Stefan C. Dentro[1,5], Matthew D. Young [1], Tim H. H. Coorens [1], Hyunchul Jung[1], Tim Butler [1], Matthew D. C. Neville [1], Daniel Leongamornlert [1], Mathijs A. Sanders[1,6], Yvette Hooks[1], Alex Cagan [1], Thomas J. Mitchell [1,2], Isidro Cortes-Ciriano [5], Anne Y. Warren[2], David C. Wedge [3,7], Rakesh Heer[8,9], Nicholas Coleman[2,10], Matthew J. Murray [2,10], Peter J. Campbell [1], Raheleh Rahbari [1,13✉] & Sam Behjati [1,2,11,13✉]

Germ cell tumours (GCTs) are a collection of benign and malignant neoplasms derived from primordial germ cells. They are uniquely able to recapitulate embryonic and extraembryonic tissues, which carries prognostic and therapeutic significance. The developmental pathways underpinning GCT initiation and histogenesis are incompletely understood. Here, we study the relationship of histogenesis and clonal diversification in GCTs by analysing the genomes and transcriptomes of 547 microdissected histological units. We find no correlation between genomic and histological heterogeneity. However, we identify unifying features including the retention of fetal developmental transcripts across tissues, expression changes on chromosome 12p, and a conserved somatic evolutionary sequence of whole genome duplication followed by clonal diversification. While this pattern is preserved across all GCTs, the developmental timing of the duplication varies between prepubertal and postpubertal cases. In addition, tumours of younger children exhibit distinct substitution signatures which may lend themselves as potential biomarkers for risk stratification. Our findings portray the extensive diversification of GCT tissues and genetic subclones as randomly distributed, while identifying overarching transcriptional and genomic features.

[1] Wellcome Sanger Institute, Hinxton, UK. [2] Cambridge University Hospitals NHS Foundation Trust, Cambridge, UK. [3] Big Data Institute, Nuffield Department of Medicine, University of Oxford, Oxford, UK. [4] MRC Molecular Haematology Unit, Weatherall Institute of Molecular Medicine, University of Oxford, Oxford, UK. [5] European Molecular Biology Laboratory, European Bioinformatics Institute (EMBL-EBI), Cambridge, UK. [6] Department of Hematology, Erasmus University Medical Center, Rotterdam, The Netherlands. [7] Manchester Cancer Research Centre, Division of Cancer Sciences, University of Manchester, Manchester, UK. [8] Translational and Clinical Research Institute, Faculty of Medical Sciences, Newcastle University, Newcastle upon Tyne, UK. [9] Newcastle Urology, Freeman Hospital, Newcastle upon Tyne Hospitals NHS Foundation Trust, Newcastle upon Tyne, UK. [10] Department of Pathology, University of Cambridge, Cambridge, UK. [11] Department of Paediatrics, University of Cambridge, Cambridge, UK. [12] These authors contributed equally: Thomas R. W. Oliver, Lia Chappell. [13] These authors jointly supervised this work: Raheleh Rahbari, Sam Behjati. ✉email: rr11@sanger.ac.uk; sb31@sanger.ac.uk

Germ cell tumours (GCTs) encompass a diverse spectrum of benign and malignant neoplasms of primordial germ cell (PGC) origin. Most malignant cases occur in the gonads of males aged 15–44 years old[1]. These tumours can be broadly divided by their histological composition into pure seminomas and nonseminomatous GCTs (NSGCTs). The latter, uniquely, can resemble a variety of tissues of embryonic and extraembryonic origin and are frequently composed of multiple histologies at once[2]. The histological composition of GCTs is of central significance for prognosis and patient management, with nonseminomatous GCTs conferring a worse prognosis[3,4]. While regional genetic changes appear to underpin histological diversification in some cancers (e.g. the childhood kidney cancer, Wilms tumour[5]), no such evidence exists to support this theory in NSGCTs to date. Furthermore, the extent to which histogenesis in GCTs mirrors normal tissue development has not been established.

These questions may be answered by an investigation of GCTs that combines both genomic and transcriptional readouts. Previous efforts in this regard either utilised bulk sequencing of tissues, where tissue-specific information within tumours containing mixed histological components would have been lost, or studied microdissected tissue through limited assays such as targeted DNA sequencing[6–11]. The latter approach was guided by variants found in a bulk whole genome taken from the primary tumour; an approach that likely lacks the necessary resolution to resolve tumour subclonality in detail[9]. Nevertheless, these efforts and others have yielded key insights about postpubertal testicular GCTs; they undergo whole genome duplication to form germ cell neoplasia in situ followed by the acquisition of further 12p copies to facilitate invasion[12].

Here, we examine the whole genomes and transcriptomes of 14 distinct histologies across 22 GCTs and four background normal testes at the resolution of individual histological units, to study the interplay of genetic diversification and tissue differentiation. We find no correlation between phylogeny and histogenesis in NSGCTs, although a conserved pattern of early whole genome duplication and later clonal diversification is observed. Each postpubertal GCT tissue retains fetal lineage-defining transcriptional expression and overexpression of the genes lying on the ubiquitously gained chromosome 12p. Prepubertal yolk sac tumours, in contrast, possess distinct genomic features such as a later whole genome duplication event and specific mutational signatures. These features represent putative biomarkers that may have a role in risk stratification.

## Results

**Overview of study**. We assembled a primary cohort comprising 15 GCTs, encompassing 11 NSGCTs, three pure seminomas and one case of a testicular prepubertal yolk sac tumour (median age 27 years, range 1–58) (Supplementary Fig. 1). All samples were taken from the primary tumour. One case had only in situ disease available for analysis. Using laser capture microdissection, we excised 547 distinct histological units (DNA and RNA in 12/15 cases, DNA only in one case and RNA only in two; Fig. 1a, Supplementary Data 1). This included 131 microdissections from tumours for DNA and 353 for RNA, as well as 63 from four regions of normal testis that served as a reference for the RNA experiments. In addition, we studied a further seven cases of prepubertal and peripubertal yolk sac tumours (age range 0.75–12 years, Supplementary Data 1) by orthogonal bulk whole genome sequencing (WGS), to validate findings from the primary cohort.

**The somatic landscape of GCT microdissections**. We generated WGS for 131 of the microdissected histological units, comprising 13 GCTs (Fig. 1a, Supplementary Data 1). The median coverage per tumour microdissection was 30 (range 15–48) (Supplementary Data 2). All classes of somatic changes were called through an extensively validated variant calling pipeline (Methods, Supplementary Data 2–6). We calculated the median cancer cell fraction of substitutions called per dissection, corrected for copy number, to be 0.96 (range 0.81–1.08) (Supplementary Data 2, Supplementary Fig. 2), indicating that these microbiopsies largely represented monoclonal tumour cell clusters.

In terms of overall mutation burden and driver variants, the GCT microbiopsies broadly matched previous reports on postpubertal GCTs (Fig. 1b). In keeping with reports from exome analyses[6,8], genomes of invasive GCTs exhibited 0.49 substitutions and 0.03 indels per Mb (Supplementary Data 2). Within tumours that harboured both in situ and invasive disease, the former harboured fewer mutations (511 vs 1324 and 893 vs 1206 median substitution burden for PD42036 and PD46966 respectively, $p < 0.001$ and $p = 0.02$, two-sided Wilcoxon rank-sum test) (Fig. 1b). We identified typical GCT driver events in our cohort, including KRAS substitutions and gains of the KRAS and KIT oncogenes, as well as other drivers such as an AKT1 substitution and a homozygous deletion of the tumour suppressor gene PTPRD (Supplementary Data 4, 5, 7, 8)[6,8,11]. Moreover, tumours exhibited typical GCT copy number profiles, as defined using 103 malignant testicular GCTs assembled by The Cancer Genome Atlas (Methods, Supplementary Fig. 3)[6]. Taken together, these findings confirmed that our primary cohort largely represented classical GCT genomes, making our downstream analyses generalisable to GCTs more widely.

One exception to this was the case of a prepubertal yolk sac tumour (PD43299), which exhibited a single base substitution (SBS) signature not typically associated with GCTs. In postpubertal GCTs, most substitutions were, as expected, attributed to signatures representing errors of cell division (SBS5/40 and, to a lesser extent, SBS1) (Methods, Supplementary Fig. 4, Fig. 1b, Supplementary Data 9) or prior treatment (i.e., the platinum agent exposure SBS35 in case PD46966). By contrast, the prepubertal yolk sac tumour predominantly harboured substitutions assigned to SBS18. SBS18 is a non-ubiquitous signature that is most prevalent in normal human placental tissue and in the childhood cancers neuroblastoma and rhabdomyosarcoma[13,14]. We pursued this finding through an orthogonal sequencing approach, using bulk tissues in an extended cohort of seven pre- and peripubertal testicular, ovarian and extragonadal yolk sac tumours. The results confirmed that pre- and peripubertal yolk sac tumours were distinguished by a contribution of SBS18. In addition, we discovered three further signatures heavily enriched in these cases; SBS17a, SBS17b and SBS-A. SBS-A is an undescribed signature characterised by C > G substitutions in an A[C > G]G trinucleotide context (signature A in Fig. 1c). Of note, the latter signature was absent from a recent meta-analysis of 23,829 cancer samples[15]. The distinct profile of these cases was further underlined by the significantly higher burden of structural variants, including retrotransposition events, in patients aged 0–12 years vs those aged 18 or older (median values of 36.5 vs 23.3 and 16.5 vs 0 for non-retrotransposition and retrotransposition events respectively, $p = 0.001$, two-sided Wilcoxon rank-sum test) (Supplementary Fig. 5). Together, these genomic features indicated that mutational processes underpinning GCT formation may vary across age groups.

**Phylogeny and clonal diversification of tumours**. To study the interplay between histogenesis and clonal diversification, we built phylogenetic trees for each microdissected tumour using somatic mutations (Methods, Supplementary Fig. 6, Supplementary

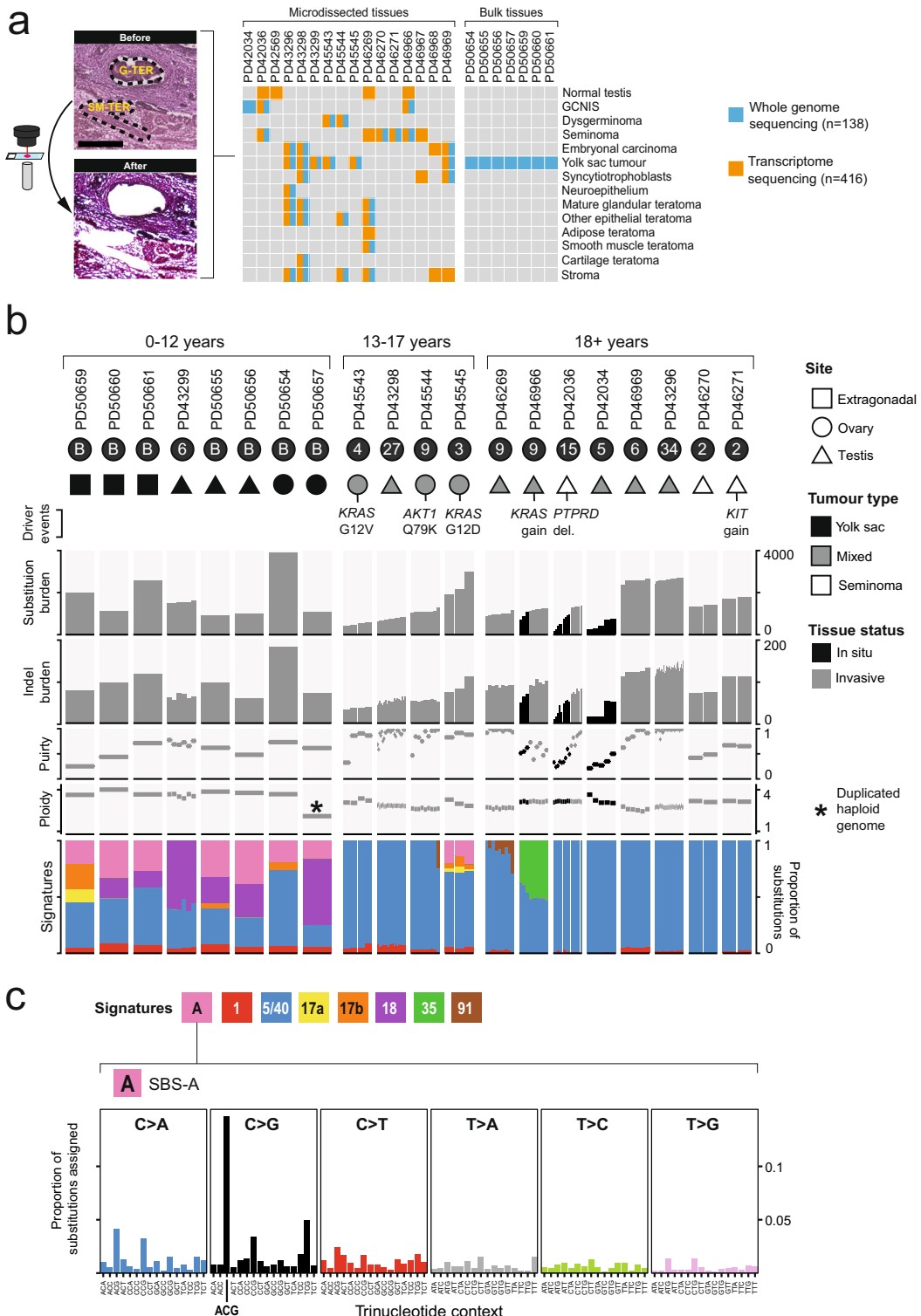

**Fig. 1 The mutational profile of GCTs. a** Overview of the experimental design, including micrographs from PD46269 illustrating different tumour histologies amenable to microdissection and sequencing (Supplementary Data 1). Gamma changes have been applied to these micrographs to help distinguish the histological features. G-TER, glandular teratoma; SM-TER, smooth muscle teratoma. Scale bar represents 250 microns. The total number of whole genomes includes both the microdissected (131) and bulk (7) samples. Note that for the mixed tumours PD42034 and PD45545 only one histology was available for microdissection and PD42569 was normal testis from a healthy donor. **b** Summary plot of key genomic data and relevant metadata pertaining to each GCT analysed (Supplementary Data 1–2, 9). Samples are ordered by patient and age. Each patient is labelled as having either a single bulk whole genome (B) or with the number of individual histological units microdissected for WGS (dark grey circle beneath each patient ID). See Methods for driver annotation. **c** Trinucleotide context plot of SBS-A. Source data are provided as a Source Data file.

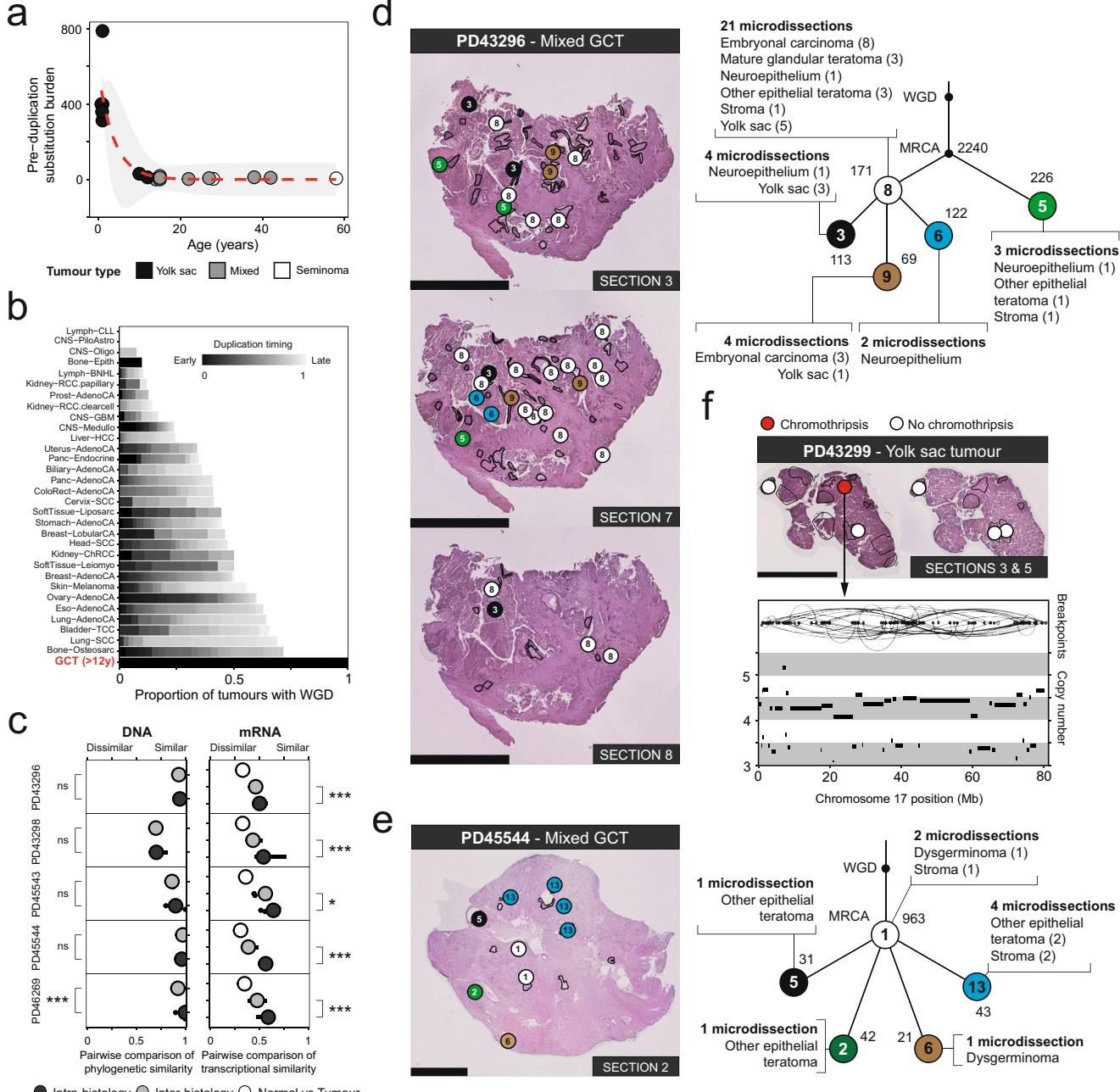

Data 10). Our analyses were based on the principles that (i) mutations shared between different tumour regions define a common phylogenetic trunk and that (ii) copy number gains defining the trunk can be timed relative to the acquisition of point mutations. For example, mutations that occurred prior to duplication of an allele would be present on both derivatives, whereas variants generated after duplication would be confined to one copy only.

Overall, GCT formation across histologies and individuals was characterised by several recurrent features. Firstly, truncal mutations included all identified driver events and represented the majority of mutations found in each invasive tumour, suggesting subclonal diversification was a relatively late event (Supplementary Data 11). Most copy number gains occurred within this trunk at a similar mutation time, consistent with whole genome duplication (Methods, Supplementary Fig. 7). The duplication itself universally arose very early, with only

~5.8 substitutions estimated to occur prior to this event across the entire genome of postpubertal (>12 years) GCTs (range 0–18) (Methods, Fig. 2a, Supplementary Data 12). This makes postpubertal cases most unusual when compared with 2096 cancer whole genomes across 31 tumour types analysed by the Pan-Cancer Analysis of Whole Genomes[16] (Fig. 2b, Supplementary Data 13). In contrast, we estimated the median pre-duplication burden in patients aged 0–12 years to be 359.7 substitutions (range 10.6–788.8), suggesting that whole genome duplication, while a universal feature of GCT, occurs relatively late in the development in malignant tumours of the young. No convincing evidence was found to indicate that the driver missense variants occurred prior to duplication (Supplementary Data 7).

We next assessed when precisely during PGC development whole genome duplication occurred by dividing the pre-duplication substitution burden estimates by the reported

**Fig. 2 Whole genome duplication timing and tumour diversification. a** Estimated burden of substitutions across the genome prior to the duplication event by age (Supplementary Data 12). Dashed red line is the fitted asymptotic regression with the grey ribbon indicating the 95% confidence intervals. The underlying equation is: pre-duplication substitution burden $= -0.59 + 574.02 * e^{(-0.26 * age)}$. **b** Bar plot comparing the prevalence and timing of WGD between our postpubertal GCTs ($n = 10$ tumours) and the tumours analysed by PCAWG (Supplementary Data 13)[16]. Tumour abbreviations used are as per the PCAWG studies (see Source data). **c** Pairwise comparison of the genetic and transcriptomic similarity of microdissections within GCTs where multiple tissues underwent DNA and mRNA sequencing (Supplementary Data 10) - PD43296 ($n = 34$ genomes, $n = 60$ transcriptomes), PD43298 ($n = 26$ genomes, $n = 48$ transcriptomes), PD45543 ($n = 4$ genomes, $n = 6$ transcriptomes), PD45544 ($n = 9$ genomes, $n = 24$ transcriptomes), PD46269 ($n = 9$ genomes, $n = 37$ transcriptomes). The $p$-values for the one-sided permutation test using label-swapping, comparing each tumour's intra- and inter-histology genomic similarity are 0.355, 0.720, 0.429, 0.257, and <0.001 respectively. Using the same statistical test for the assessment of transcriptomic similarity, the $p$-values are <0.001, <0.001, 0.015, <0.001 and <0.001. The results are uncorrected for multiple hypothesis testing. Annotations for $p$-values: ns, not significant; *<0.05; ***<0.001. Where $p$-values are <0.001, no random permutation of the data for 1000 re-samples captures as large a difference between the two groups. Source data are provided as a Source Data file. **d, e** Histological images from example testicular **d** and ovarian **e** mixed GCTs that underwent extensive multiregional sampling, each annotated with the mutation clusters that define the phylogenetic relationship of each microbiopsy (Supplementary Data 10). Circles on the histological images correspond to a numbered mutation cluster in the associated phylogeny on the right-hand side. The number next to each cluster denotes the number of autosomal substitutions that support it. Circles are coloured to spatially highlight the clonal composition of each tumour. Each cluster is labelled with the number of microdissections for which it is the major clone and a list of the histologies the cluster pervades. MRCA, most recent common ancestor. These figures are simplified versions of the full phylogenies (Supplementary Fig. 6). **f** Subclonal chromothripsis of chromosome 17 in a prepubertal yolk sac tumour (Supplementary Data 4, 5). Reconstructed breakpoints are illustrated above the copy number calls. Scale bars represent 2.5 mm.

mutation rate per cell division within PGCs[17]. We estimated the median postpubertal GCT whole genome duplication occurred at ~5 cell divisions post-PGC specification (range 0–15) (Methods), overall placing the genetic hallmark of GCT initiation—whole genome duplication—in fetal life for many tumours.

**Relation of histogenesis to clonal diversification.** Next, we asked whether histogenesis and genetic diversification were correlated by analysing both DNA and RNA from the same histological units. We generated cDNA libraries through a modified single cell mRNA sequencing protocol (Methods). A total of 112 out of 131 whole genomes derived from microdissections had transcriptomic data from adjacent regions available (Supplementary Data 2). These libraries yielded expression profiles for 55,502 non-mitochondrial features with a median 298,080 reads mapped per microbiopsy (range 13,822–3,863,511 reads) (Supplementary Data 14). We performed pairwise comparisons to identify the similarities of genomes and transcriptional profiles within each GCT (Methods, Fig. 2c). We considered comparisons of the same and of different histologies within each tumour. The results suggested that biopsies of the same histology were not necessarily more closely related than two biopsies of different tissues in most cases. Instead, genomic diversification followed an anatomical pattern where, within each patch of a unique somatic subclone, various histologies had been generated, as our reconstructed phylogenies elucidated (Fig. 2d, e, Supplementary Fig. 6, Supplementary Data 10). By contrast, the transcriptional expression within a single GCT histology was significantly more similar compared with other tissues, indicating that histology-specific, protein-coding transcription transcended genetic heterogeneity (Fig. 2c). An unusual example was seen in a pure yolk sac tumour in which one subclone was defined by 66 structural variants rearranging chromosome 17 (chromothripsis) which generated potential driver events (e.g., loss of *TP53*), as previously described in an osteosarcoma (Fig. 2f)[18]. However, chromothripsis did not significantly perturb the global transcriptional proximity between different tumour clones ($p = 0.89$, one-sided permutation test) (Methods, Supplementary Fig. 8). In aggregate, our findings indicate that histogenesis in GCTs is not governed by somatic genetic diversification. This finding corroborates a previous observation in bulk data of transcriptional clustering by gross histological category despite the absence of an apparent unifying genomic event[6].

**Fetal signals underpin GCT histogenesis.** A fundamental question of GCT histogenesis is whether differentiated tissues, such as a region of GCT cartilage, show abnormal gene expression relative to their adult, normal counterparts. We addressed this question by comparing transcriptomes of GCT tissues with reference transcriptomes of corresponding fetal and mature tissues, as defined by single cell mRNA sequencing. We examined a total of 416 histological units from 14 tumours and four regions of histologically normal testes covering a total of 14 histological structures (Fig. 1a, Supplementary Fig. 1, Supplementary Data 1).

Using single cell reference data from fetal and adult tissues corresponding to each GCT tissue[19–23], we found that tumours not only expressed lineage-specific transcripts but also consistently retained expression of fetus-specific features (Fig. 3a–c). This remained the case even in apparently mature tissues such as the smooth muscle teratoma where *IGF2*, typically restricted to high expression in the fetus[24], was readily detectable alongside the typical smooth muscle markers *ACTA2*, *MYH11* and *TAGLN*[23]. In contrast, the microdissected adult seminiferous tubules did not demonstrate the fetal transcriptional signal seen in the malignant tissues recapitulating primordial germ cells, i.e. seminoma, dysgerminoma and GCNIS (Fig. 3a).

If the transcriptomes of GCT tissues resembled their fetal correlates, we might then better appreciate this by performing differential expression analysis between each individual NSGCT component and the NSGCT subtype embryonal carcinoma, which phenotypically recapitulates human embryonic stem cells (hESCs). (Methods, Supplementary Data 15)[25,26]. Here, illustrated through use of hESCs markers, as well as those corresponding to each differentiated tissue, we found gradients of relative expression akin to those seen in embryogenesis (Fig. 3d–f). Maturing tumour tissues overexpressed genes known to regulate cell fate specification of their matching fetal tissues, such as in neuroepithelium (*SOX1*, *SOX3* and *PAX6*) and smooth muscle (*MYOCD*, *MIR143HG* and *MIR145*)[27–30]. This lineage specification is likely to be determined at an epigenetic level, as has been suggested before[31].

**A canonical GCT transcriptome.** Despite the histological and protein-coding transcriptional diversity of GCTs, it is conceivable that there is a global component of the transcriptome that transcends tissues and tumours. We found significant enrichment of chromosome 12p gene expression across the invasive tumour histologies (Fig. 4a, Supplementary Data 16), by assessing the

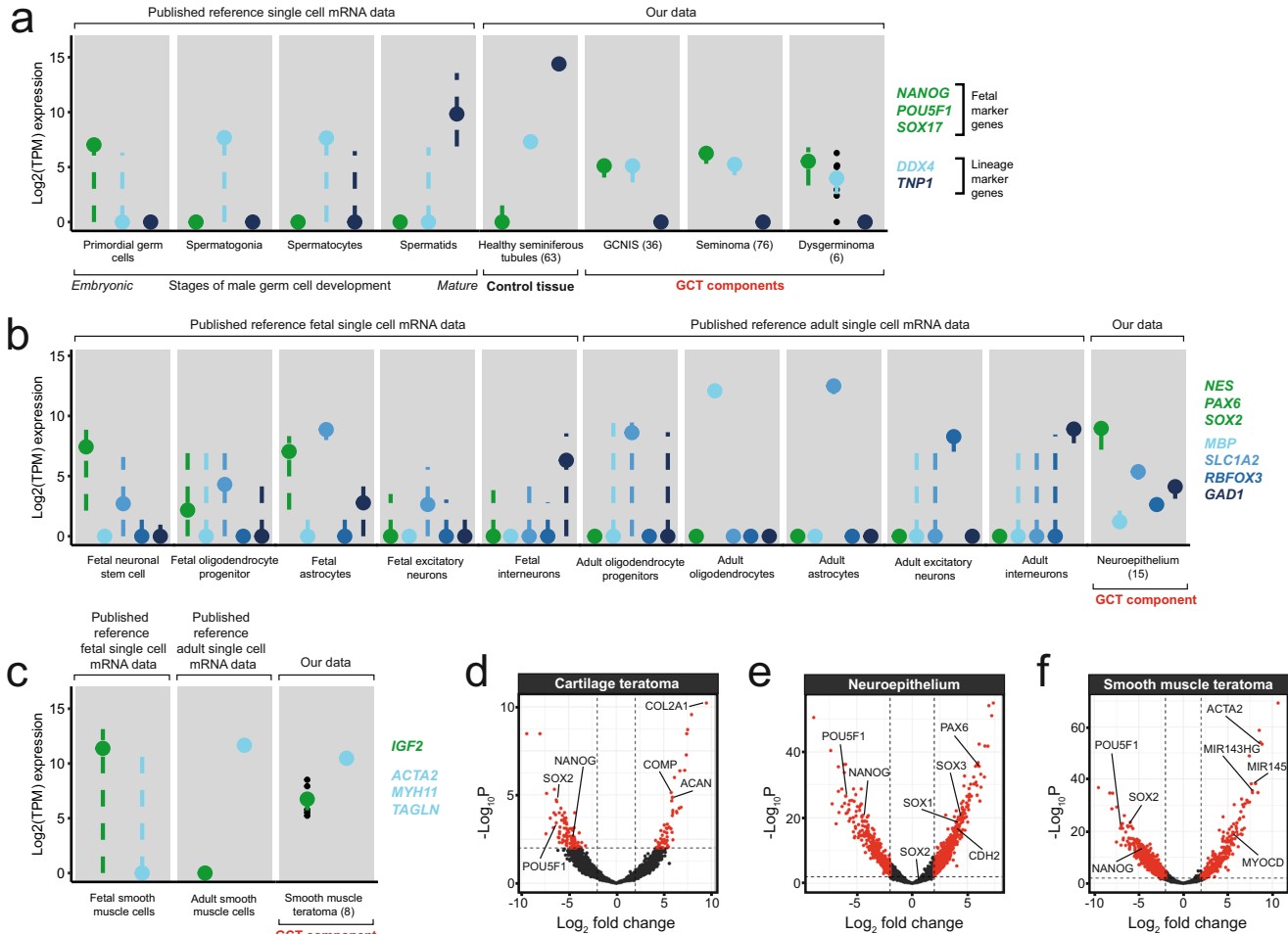

**Fig. 3 Pathways of GCT histogenesis. a–c** Comparison of the expression of marker genes in example GCT tissues and their corresponding fetal and adult counterparts, using reference single cell datasets[19-23]. Coloured dots denote the median and dashed lines the interquartile range for the underlying data points which represent the expression of a single gene per microbiopsy per tissue. Individual data points are plotted where the total data points supporting a point range are ≤10. The total number of microbiopsies supporting the data in each grey panel is provided in brackets within the figure. Source data are deposited on Mendeley. **d–f** Example volcano plots illustrating the differential expression analysis between embryonal carcinoma and more mature GCT tissues. The dashed lines represent the cut-offs for the log2 fold-change (>|2|) and adjusted $p$-value (<0.01, Benjamini-Hochberg correction) considered significant. Genes enriched in each differentiated tissue are shifted to the right. Marker genes used to define embryonic stem cells and each differentiated tissue are annotated on the plot. The full list of genes and tissues comparisons can be found in Supplementary Data 15.

expression profile of each GCT tissue, relative to normal testis within each cytoband along the genome (Methods). The enrichment corresponded to copy number gain, with 12p genes generally possessing a significantly higher log2 fold-change relative to healthy seminiferous tubules than regions nearer baseline ploidy ($p < 10^{-5}$, one-sided permutation test) (Methods, Fig. 4b, Supplementary Fig. 9). This finding was of particular importance and plausibility as gains of chromosome 12p (usually arranged in an isochromosome) are a near universal feature of postpubertal GCTs and is consistent with previous bulk sequencing of GCTs and other malignancies[32,33]. We note that the mapping of the transcriptional change to copy number was not exact and may be explained in part by use of relative expression to normal testis and the influence of gene promoters that lie outside of the region gained. Examining the expression of individual genes along 12p revealed the bona fide oncogene *CCND2* to be universally overexpressed in all invasive tumour histologies investigated using our conservative cut-offs (≥2 log2 fold-change and <0.01 adjusted $p$-value), as well as *ATN1* and *PTMS*. *KRAS*, in contrast, was only overexpressed in embryonal carcinoma (Supplementary Data 17). PTMS (parathymosin) is thought to facilitate chromatin

remodelling through interaction with the linker histone H1 and can induce human sperm nuclei to undergo decondensation, possibly implicating it in GCT epigenetic remodelling[34]. Other features that were widely expressed (*SLC2A3*, *PHB2*) were consistent with previous reports[32]. Together, a picture emerged confirming a conserved GCT transcriptome in the form of 12p gene overexpression, regardless of invasive histology, driven by the defining 12p gain.

## Discussion

We studied the origins and tissue diversification of GCTs from DNA and mRNA sequences derived from microdissected histological units. Our approach enabled us to directly overlay anatomical boundaries of clonal diversification with histological features. Our result revealed that histogenesis in GCTs is not demarcated by clonal territories. Instead, GCT tissues appear to arise independently of somatic diversification along transcriptional pathways that broadly mirror normal human tissue development. An important distinction from normal histogenesis was the retention of fetal developmental gene expression in GCT tissues, even in

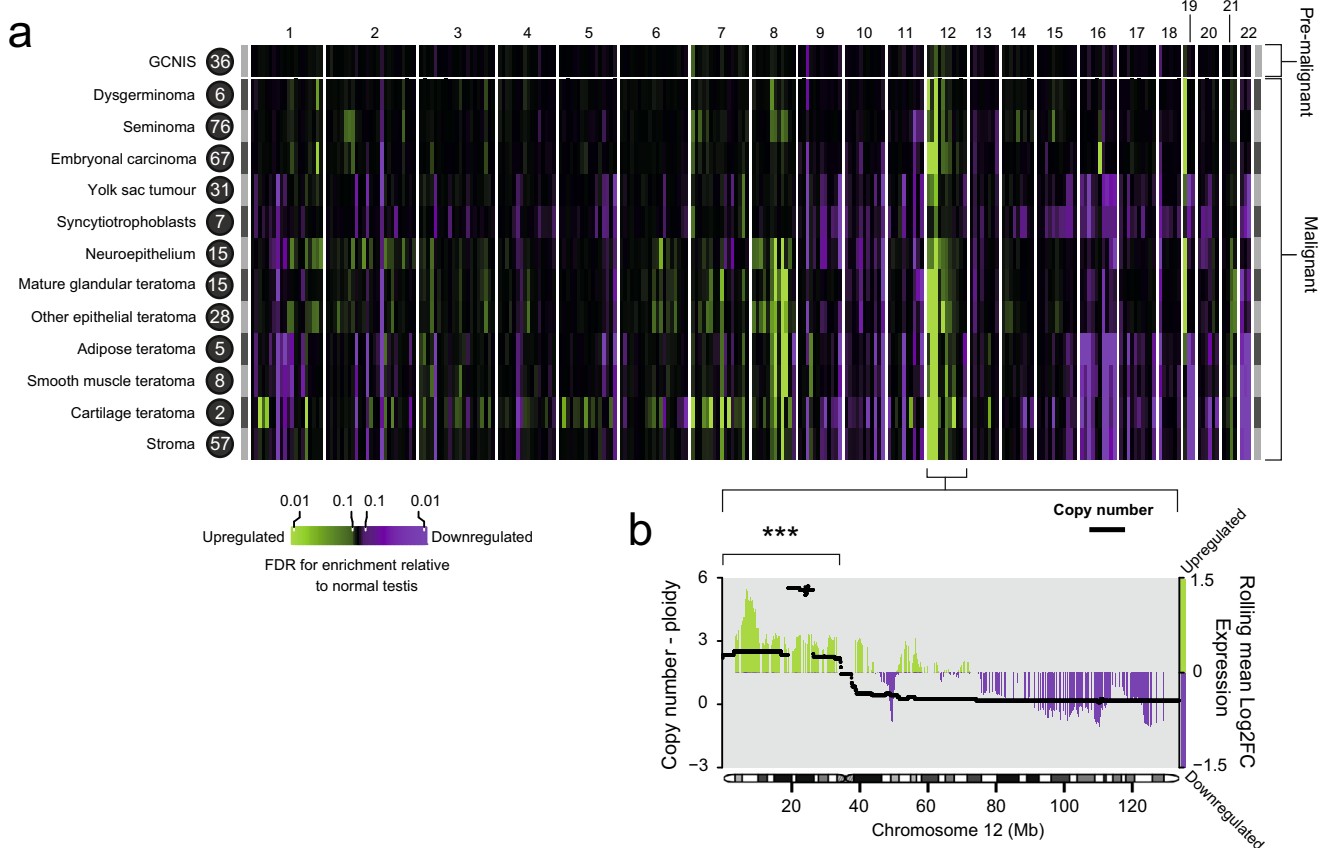

**Fig. 4 The relationship between GCT genome and transcriptome. a** Heatmap showing gene enrichment per GCT tissue relative to healthy seminiferous tubules, binned by cytoband. Colours correspond to significance of enrichment according to the adjusted *p*-value (false discovery rate correction). The number next to each histology is the number of eligible microbiopsies that informs the analysis (Supplementary Data 16). **b** Combined plot of the chromosome 12 copy number changes across all invasive tumours and the rolling average log2 fold-change in gene expression compared with healthy seminiferous tubules. The window size for the rolling average is 50 genes. The average log2 fold-change in expression across 12p was significantly higher than across comparable numbers of genes found across regions with near baseline ploidy (one-sided permutation test, $p < 1 \times 10^{-5}$) (Supplementary Fig. 9), as indicated by the three asterisks. Source data are provided as a Source Data file.

seemingly mature tissues, that emerged as a defining feature of GCT components. It is conceivable that this incomplete maturation represents a targetable vulnerability for interventions that promote differentiation, akin to therapeutic efforts in childhood cancer that aim to overcome maturation blocks[35–38]. Instances of more aggressive, differentiated GCTs are rare, such as non-gestational choriocarcinoma which may reflect the normal invasive potential of the trophoblasts they recapitulate[39]. Beyond the retention of fetal transcripts, we identified two further stereotypical features of GCT tissues. First, we observed dysregulation of gene expression on chromosome 12p, amplification of which is a key genomic hallmark of GCT, to be conserved across postpubertal GCT tissues. Second, the somatic genetic development of GCT genomes exhibited extensive diversification yet was unified by a common root that lay in WGD.

It has previously been suggested that pre- and postpubertal GCTs may arise from different stages of PGC specification[40]. The key evidence supporting this notion includes variations in the tissue composition of GCTs across different age groups, a reduced prevalence of 12p gain in tumours of the young, and nuances in translational protein-coding gene profiles[41–44]. Our investigation has revealed stark somatic genetic differences between pre- and postpubertal GCTs. While the near universal step of WGD was preserved across all GCTs, it consistently occurred later in tumours of the young although the exact mechanisms by which it occurs in each case requires further investigation. Furthermore,

our analyses revealed distinct mutational signatures that underpin many substitutions in GCTs of young children. These findings would support the view of a fundamental developmental distinction between pre- and postpubertal GCTs that may delineate a subgroup of 'true' paediatric tumours which may be of clinical relevance. A major challenge in the clinical management of GCTs remains whether to treat older, peripubertal children according to 'paediatric', less intense treatment protocols, or as 'adults' who generally require more intensive cytotoxic chemotherapy[45]. We speculate that timing of whole genome duplication and distinct mutational signature profiles may lend themselves as potential tangible biomarkers for treatment stratification in this context.

Our investigation has directly addressed the relationship between genetic and transcriptional diversity in GCTs. Although limited by the relatively few cases that possessed each single histology or many simultaneously, the high-resolution multi-omics approach of our study seems to indicate that the superficially heterogeneous GCTs still possessed canonical transcriptional and genetic features that underpin GCT development. We would expect our approach to be broadly applicable across human cancer to study the interplay of somatic changes and transcription at the level of microscopic tumour regions.

## Methods
**Tissue collection, handling and microdissection.** Informed, written consent was obtained from participants or their legal guardians. No compensation was provided

to participants. Samples were stored fresh frozen after collection in accordance with protocols approved by the local ethics committees. Samples were collected using the following UK REC approval numbers (with the committee that provided them in brackets):

- 03/018 (Cambridge Central-NRES Committee East of England)
- 08/h0405/22 + 5 (Derby-NRES Committee East Midlands)
- 12/NE/0395 (Newcastle & North Tyneside 1-NRES Committee North East)
- 16/EE/0394 (East of England—Cambridge Central Research Ethics Committee)
- 17/LO/1801 (London—Surrey Research Ethics Committee)
- 18/EM/0134 (East Midlands—Derby Research Ethics Committee)
- 18/NW/0092 (North West—Greater Manchester South Research Ethics Committee)

The tissues in the primary cohort were fixed using PAXgene fixative (Qiagen) and paraffin embedded in preparation for microdissection[46]. To ensure that each component of the tumour isolated was correctly identified, a reference H&E slide and slides stained with the relevant immunohistochemical antibodies were generated and reviewed by a consultant histopathologist (Supplementary Fig. 1).

Each sample was taken from the primary tumour. Age at diagnosis was used to categorise patients.

**DNA library preparation and sequencing**. Low-input, whole genome sequencing libraries were created from each microbiopsy[46]. 150 bp paired-end reads were then generated from these libraries using either the Illumina HiSeq XTEN or NovaSeq sequencing platforms to a target coverage of 30×. For the additional bulk samples, all seven were sequenced on the NovaSeq platform only (Supplementary Data 2). These reads were aligned to the GRCh37d5 human reference genome using the Burrows-Wheeler Aligner (BWA-MEM)[47].

**RNA library preparation and sequencing**. Microbiopsies were collected in wells pre-filled with 50 μl of RLT Plus lysis buffer (Qiagen). Regions where micro-dissections had been taken for DNA sequencing were specifically targeted, using the adjacent tissue sections. RNA microdissections were considered adjacent if they were <1 mm from where a successful DNA library had been generated and were of the same histological feature (Supplementary Data 2).

AMpure XP beads (Beckman Coulter) were used to recover RNA from the lysate. Oligos and reagents from the Smart-seq2 protocol[48] were used to reverse transcribe and amplify full-length cDNA. Libraries were prepared from the cDNA using the NEBNext Ultra II FS DNA Library Prep Kit for Illumina (New England Biolabs). Libraries were sequenced with the Illumina HiSeq 2500 or 4000 systems to produce 100 bp paired-end reads. Adaptors and low quality reads were removed using Trim Galore (https://github.com/FelixKrueger/TrimGalore) and the following parameters: -q 20–fastqc–paired–stringency 1–length 20 -e 0.1. The Spliced Transcripts Alignment to a Reference (STAR) aligner was used to map the raw sequencing reads to the GRCh37d5 human reference genome[49].

**Estimating contamination of tumour samples**. To exclude contamination as a source of false positive variant calls, we ran the Conpair algorithm[50] on all whole genome data included.

**Estimating the callable length of the tumour genome**. When calculating the mutation burden per megabase, we adjusted for the length of the genome we were able to call mutations across. We ran Mosdepth[51] to exclude loci that have fewer than 4 reads covering them and then removed regions masked by the variant callers due to their highly repetitive nature.

**Variant calling and filtering**. All DNA mutation calling was performed against a matched normal blood or non-neoplastic testis microbiopsy sample.

Substitutions were called using the CaVEMan algorithm[52]. We only kept candidate variants if the reads supporting them passed a minimum median alignment score (≥140) and fewer than half were clipped. For bulk samples, variants needed to be supported by ≥4 variant reads with a total coverage of ≥10×. Further artefacts resulting from the low input DNA library preparation method in the microdissected samples were then removed using previously validated filters[46]. Low input pipeline substitution artefacts are usually found on reads that share a similar alignment. Consequently, these filters are based on statistics such as a variant's position compared to the start of alignment and its standard deviation and median absolute deviation on the reads supporting it. To improve our sensitivity for calling low depth variants in any given microbiopsy, a particular problem for the lower purity and aneuploid in situ disease, we leveraged the multi-sampling nature of the experiment by conducting a pileup of all substitutions called across all patient samples with minimum base (25) and read mapping (30) quality thresholds. Putative mutations found at loci with universally low coverage (mean ≤10×) across a patient's samples were then removed as probable mapping artefacts. Finally, to differentiate true low depth variants from sequencing artefact, we fitted a beta-binomial model adapted from the Shearwater variant caller[53] to establish a locus-specific error rate using a reference panel of 250 unrelated, normal samples

that have been subject to the same library preparation and sequencing platforms[13,54,55]. A variant was considered real in a given sample if $p < 0.001$ after multiple-test correction (Benjamini-Hochberg method). To ensure the veracity of our variant calling from this pipeline, a subset of substitutions were visually inspected using the genome browser Jbrowse[56].

Small indels were identified using the Pindel algorithm[57]. All putative indels had to pass a minimum quality score (≥300), not fall within 1 bp of a SNP or 5 bp of an indel call found in the matched normal sample and not be a 1 bp indel at a homopolymer run of 7 bp or longer. Indels were then only considered if ≥5 variant reads supported them. For bulk samples, a total coverage at the indel site of ≥10× was required, as well as a minimum of at least one read supporting it in both directions. Indels from microbiopsies underwent similar genotyping to that described for substitutions to rescue true low depth variants. Genotyped variants were then filtered to remove those with a low average coverage (mean ≤10x) before being subjected to the aforementioned Shearwater-like approach. Due to computational constraints, the unmatched normal reference panel was reduced to 100 samples for indel filtering. Tumour variants seen at a VAF of ≥0.2 in the matched normal or across this unmatched panel were filtered out. Mutations were excluded if the combined sum of unknown and ambiguous reads at that position across a patient's samples was greater than the number of variant reads or if half or more total reads at that locus were of low quality (mapping quality < 20), as previously described[54,55]. Once more, inspection of a subset of the final indel calls in Jbrowse[56] validated this stringent filtering approach.

Battenberg[58] was used to call copy number aberrations and estimate the purity and ploidy. As Battenberg is primarily informed by the B allele frequency when determining the copy number state, very high purity tumour samples—such as those obtained by microdissection—with regions of loss of heterozygosity can be challenging to accurately call. To overcome this, Battenberg was run twice—once as a standard tumour and matched normal pair and the second time merging the tumour and normal sample BAM files to lower the tumour purity. The results from both runs were manually reviewed and subject to the DPClust algorithm[58] which determines the subclonal architecture of a sample using both the substitutions and copy number calls. The best fit according to this combined approach was kept. Where the merged BAM file Battenberg calls were kept, the purity from the original run was still used in downstream analyses. The DPClust algorithm was run for 10,000 iterations with the first 3000 dropped as burn-in.

Structural variants were detected using the BRASS algorithm[58]. False positive calls because of the low input DNA library preparation were removed using AnnotateBRASS[59] which was informed by the same 100-sample normal reference panel used in indel filtering. Further artefacts were flagged by manually reviewing the borderline calls that narrowly passed filtering and removing them. Additional filtering for the bulk samples consisted of removing any variants also called in the matched normal. The final step for both cohorts was to remove any remaining variants found within 100 bp of any retrotransposition event.

We called somatic retrotransposition events using a separate pipeline to the one detailed above. First, those with exact breakpoints were identified using the TraFic algorithm[60]. To identify further transduction events, the raw calls from the BRASS algorithm were examined to identify read clusters associated with known L1 germline sources. All putative variants were then manually reviewed. The final calls can be found in Supplementary Data 6.

**Mutational signature extraction**. De novo single base substitution (SBS) signatures were initially extracted using the hierarchical Dirichlet process (HDP version 0.1.5, https://github.com/nicolaroberts/hdp)[61]. This was done on a per patient basis, i.e. all unique substitutions across a tumour, to prevent double counting. This was run across 20 independent posterior sampling chains with 80,000 burn-ins and another 20,000 sampled iterations. The resultant signatures were deconvolved against the COSMIC version 3.2 SBS signatures using an expectation-maximisation algorithm[62]. The signature called in addition to the COSMIC SBS signatures in the initial run showed clear conflation with SBS1, likely because it co-localised to samples with higher SBS1 burdens (Supplementary Fig. 10). To clean this up, HDP was run once more using only the prior COSMIC signatures, including SBS1, found in the initial run. The resultant SBS-A signature is shown in Fig. 1c. All the components extracted at the final step can be seen in Supplementary Fig. 10. Signatures were then mapped back to individual samples using SigProfiler (version 1.015)[15]. SigProfiler performed an independent de novo signature extraction whose output was decomposed using the final list of HDP signatures. For the SigProfiler de novo extraction step, sigProfilerExtractor was run using a minimum and maximum of 2 and 15 signatures respectively with all other parameters set as default.

**Driver mutation analysis**. Driver mutations were initially considered in known cancer genes, as defined by the COSMIC version 94 cancer genes consensus. Missense mutations and in frame indels that occurred within genes annotated as oncogenes, or who act in a dominant manner, and were found at previously documented hotspots were considered driver events. For recessive cancer genes, all intact gene copies had to be lost to consider it a driver event. This included substitutions, small indels, structural variant breakpoints that truncated the gene footprint and copy number changes. We defined focal amplification and deletion driver events according to previously outlined heuristics[58]. Briefly, gained segments

had to be <1 Mb in size and a minimum of five or nine intact copies of the oncogene had to be gained, depending on whether the average ploidy was below or above 2.7. Focal deletions shared the same segment size and ploidy cut-offs but with a total copy number of zero or less than average ploidy minus 2.7.

The KRAS and KIT gains noted in Fig. 1b did not fulfil this definition for focal amplification, hence their notation as gain, rather than amplification. Both possessed the minimum number of copies, however they lay on segments between 1 and 10 Mb in size. They were retained and noted, however, in view of their role as classic GCT drivers.

As a second, systematic check for drivers, we employed two computational methods. The first, dNdS, tested for genes under positive selection[63]. This did not yield any significantly enriched genes, likely due to the modest cohort size. Furthermore, the CHASMplus algorithm[64] was then run to systematically identify driver substitutions across the cohort with the three previously identified KRAS and AKT1 mutations scoring most highly (Supplementary Data 8).

**Aggregated copy number comparison against TCGA data**. Using only GCT microbiopsies from invasive tissues, i.e. not GCNIS, with a purity >40%, the major clone copy number state (as defined by Battenberg) was extracted per 10 kb genome bin. The purity threshold was used to ensure confidence in the copy number calls. The median total copy number across all microbiopsies was used to generate per tumour copy number profiles from which the median tumour ploidy was subtracted to distinguish additional gains from their higher starting copy number state baseline because of WGD. The average of this value was used in a cohort-level, aggregated copy number profile. The prepubertal case (PD43299) was excluded from this analysis in view of its higher overall ploidy and lack of comparable samples in TCGA reference[6].

For the comparison with TCGA data[6], we only included TCGA samples with a purity greater than 40%, leaving 103 eligible tumours.

**Detection of chromothripsis**. Chromothripsis-like events were detected using ShatterSheek[65]. Putative calls, both high and low confidence, were manually reviewed to remove false positives.

**Tumour phylogeny reconstruction**. Tree building was performed using only samples with an average reads per chromosome copy of ≥5. The average reads per chromosome copy in a given sample was calculated using the following formula[16]:

$$\text{Average\_reads\_per\_chromosome\_copy} = \text{purity}/((\text{purity} * \text{ploidy}) + ((1 - \text{purity}) * 2)) \\ * \text{tumour\_coverage} \tag{1}$$

The approach to phylogenetic tree reconstruction depended upon the number of samples available from a given tumour. For five samples or fewer, we used the DPClust algorithm[58] with the Gibbs sampler run for 3000 iterations and the first 1000 dropped as burn-in. More highly sampled tumours were subject to multidimensional DPClust instead[66]. This iteratively compares the phylogenetic distance of sample triplets before combining all outputs into a single unifying tree structure. Due to computational constraints the iterations and burn-ins used by the Gibbs sampler here were adjusted to 1000 and 200 respectively. Both methods were informed by both the substitutions and copy number aberrations. Clusters accounting for <1% of total substitutions, fewer than 20 substitutions or small clusters that violated the remainder of the tree were removed (e.g. residual sequencing artefact that was found at low VAF across all mutation clusters) (Supplementary Data 10). Only autosomal substitutions were included in this analysis.

**Estimating the proportion of substitutions within the tumour's phylogenetic trunk**. The tumour phylogenies generated served as the input to our calculation of the proportion of substitutions found within the phylogenetic trunk. This approach had the advantage of accounting for substitutions that may be apparently absent in a given sample but were in fact lost through a subsequent deletion or chromosomal loss. The proportion was calculated using the following formula:

$$\text{Proportion\_subs\_in\_trunk} = \text{n\_subs\_in\_trunk}/(\text{n\_subs\_in\_trunk} + (\text{sum}(\text{subclonal\_branch\_length}) \\ /\text{n\_subclonal\_branches})) \tag{2}$$

For tumours with GCNIS, the mutations shared between GCNIS and the invasive tumour were included in the trunk. To calculate the average branch length from the trunk, the summed mutation distance between the trunk and each branch tip was divided by the total number of branch tips. This meant double-counting some mutations in cases where two or more branch tips shared mutations not found within the phylogenetic trunk to normalise the lengths of all subclonal branches emanating from the trunk equally (Supplementary Data 11).

**Timing copy number events**. To ensure each copy of a chromosome within a sample had sufficient coverage to accurately time clonal copy number changes along it, we limited our analysis to the 96 microbiopsies and 6 bulk samples with a calculated average of ≥7 reads per chromosome copy. We then ran

mutationtimeR[16] to identify clonal substitutions and estimate the probability that they each occurred prior to a copy gain. All analysed samples were determined to have undergone whole genome duplication as their ploidy (weighted by subclonality) was ≥2.9–2 * (Extent of homozygosity, weighted by subclonality), the approach taken by the PCAWG Consortium[16].

To estimate the substitution burden prior to WGD, we first identified all 2 + 0, 2 + 1 and 2 + 2 copy number segments that were predicted to be involved in WGD. We excluded low confidence segments, defined arbitrarily as those with ≥0.5 width confidence intervals for the mutation time estimate of their gain. All substitutions predicted to occur prior to duplication across these segments were then added together and adjusted for the copy number configuration of the tumour:

$$\text{CN\_adjusted\_burden} = \text{num\_pre\_dup\_subs} + \text{num\_pre\_dup\_subs} \\ * (\text{num\_total\_subs}_{2+1,2+0}/\text{num\_total\_subs}_{2+2,2+1,2+0}) \tag{3}$$

Num_pre_dup_subs represents our raw pre-duplication substitution counts while num_total_subs is all substitutions called across the eligible segments matching the configurations listed in the subscript. This adjustment accounts for the pre-duplication substitutions on the minor allele in regions of 2 + 0 or 2 + 1 that cannot be identified. Lastly, we extrapolated this adjusted value to a genome-wide estimate, according to how many bases were covered by our included copy number segments.

To convert the substitution burden to a number of cell divisions, we doubled a published estimate of 0.5–0.7 substitutions incurred per haploid genome per cell division within PGCs[17] to derive a per diploid genome estimate. We anticipated that, by running our variant calling against a matched normal sample, the earliest embryonic mutations (typically 1–2)[13,67,68] would not be detectable by our pipeline and thus most detectable mutations would have emerged post-PGC specification.

We considered alternative timing methods too, such as placing WGD in chronological time by restricting analyses to C > T substitutions in a CpG dinucleotide context, as was done in PCAWG[16]. The reasoning here was that these mutations are thought to accrue in a linear, clock-like fashion while other mutational processes will fluctuate over the course of a tumour's life, making real-time estimations of copy number gain events from mutation time more difficult. We deemed this approach unsuitable for our study cohort, however. GCTs are characterised by low tumour mutation burden, meaning that restricting analyses to such a limited mutation context would leave very few mutations to analyse. Perhaps more importantly though, GCNIS and seminoma are characterised by global hypomethylation which may be re-established in NSGCTs[6,69]. This is likely to influence the mutation rate at CpG sites considerably as mutations in this context are usually the consequence of spontaneous deamination of 5-methylcytosine, typically causing SBS1-pattern mutations[15]. With such dynamic changes in the methylation of the GCT genome, it cannot be assumed that the mutation rate at these sites in GCTs is linear.

Only tumour types with 10 or more samples were considered within the PCAWG dataset[16] when comparing it to our postpubertal GCT cohort.

**RNA sequencing data pre-processing**. 500 microdissections were initially taken from across the 14 tumours and 4 regions of histologically normal testis. 40 were excluded due to contamination by other tissues during microdissection, ambiguity over their histological categorisation, or because they were neither neoplastic nor seminiferous tubules (e.g. populations of Leydig cells or lymphocytic infiltrates). The library depths achieved across the 55,502 initially mapped nuclear features with a median of 246,980 mapped reads (range 682–3,863,511, interquartile range 86,612–651,605) (Supplementary Fig. 11). Reads were counted per feature using featureCounts (version 1.5.1)[70].

To filter out failed or low-quality libraries, we considered features to be expressed if at least five reads were mapped to them and assessed how many genes were expressed per microbiopsy (Supplementary Fig. 12). The median number of features expressed to this depth per sample was 8916 (range 0–22,865, interquartile range 4150–13,654). After removing 44 microbiopsies which expressed fewer than 1000 features to this depth, 416 were left for downstream analyses.

To assess the transcriptional relationship between microbiopsies, we inspected a uniform manifold approximation and projection (UMAP) of their transcriptomes using the Seurat package[71] (Supplementary Fig. 13). Histology was a strong determinant of sample clustering although we found embryonal carcinoma in particular retained patient-specific signals. We found no evidence of batch effect. We used previously described marker genes from common GCT histologies to further assess the data quality and found their expression to localise to the expected tissues[72–75] (Supplementary Fig. 14).

**Differential expression analyses**. The Limma-Voom method[76] was used to perform all differential expression analyses. The default parameters were used to filter genes by a minimum expression within a test group necessary for differential expression to be confidently determined.

To identify the significantly enriched and depleted regions of gene expression across each GCT histology isolated, differential expression analysis was performed between each tissue and the healthy seminiferous tubule reference microbiopsies. Changes in expression at the level of the cytoband could then be confirmed by gene enrichment analysis using Limma's 'camera' function and the MSigDB C1 gene set[77].

A pan-GCT expression profile for chromosome 12 was constructed by performing differential expression analysis on all invasive GCT tissues together against healthy seminiferous tubules, with the histological subtype set as the blocking factor (random effect) to adjust for histology-specific variability in 12p gene expression. In order to establish whether the high expression across 12p—which was universally gained in tumours that had undergone both DNA and RNA sequencing—could be due to chance, we performed 100,000 random samples of genes from genomic regions near baseline ploidy and measured their average log2 fold-change in expression and compared them to the average across 12p. 'Near baseline' was arbitrarily set as ±0.5 away from the Battenberg ploidy estimate. Each sample contained 226 genes, reflecting the 226 genes that lay on 12p which were retained during the pan-GCT differential expression analysis. The remainder were filtered out due to a paucity of mapped reads in both the normal testis and GCT samples.

**Pairwise genetic and transcriptomic similarity scores**. For tumours where DNA and mRNA sequencing for at least 2 invasive histologies were available we devised scores to indicate how similar a given pair of microdissections were. The genetic score was derived from the proportion of substitutions that the two cuts share, compared to their average burden, as determined by the mutation clusters called across each tumour (see Tumour phylogeny reconstruction). A mutation cluster was considered present in a sample if its cancer cell fraction was >0.1.

For the transcriptomic similarity, we compared only protein-coding genes and excluded gene sets that were not of interest or likely to add noise to the analysis, including haemoglobin, immunoglobulin, cycling and housekeeping genes. Subsequently, any given two samples were compared by measuring the Pearson correlation coefficient of the log2(TPM) values between them. PD46969, although possessing both DNA and mRNA for multiple invasive GCT histologies, was excluded from this analysis on the basis that multiple samples were available for microdissection but the tissues isolated were not well represented across all biopsies. For example, yolk sac tumour DNA sequences were all derived from one biopsy and syncytiotrophoblasts from the other. Such large spatial biases were likely to confound assessment of the relationship between histology and phylogeny.

To assess the significance of the differences in genomic and transcriptomic similarity we observe between the intra- and inter-histological scores, we randomly swapped the labels for all pairwise comparisons in a tumour 1000 times and plotted the difference in the median scores generated. This permutation approach provided a distribution from which a $p$-value could be derived. For PD45543, which only had four whole genomes, 15 combinations would have exhausted the random sampling space, so we compared the observed genomic difference to the 14 possible alternatives.

**Global transcriptional effects of chromothripsis**. To assess whether subclonal chromothripsis of chromosome 17 in PD43299, a pure yolk sac tumour, had a significant impact on the transcriptome, we examined the intra-tumoral Pearson correlation between regions of yolk sac tumour. 6 tumours were included. Our hypothesis was that chromothripsis would increase transcriptional heterogeneity which could be measured through an increase in variance. After calculating the variance of yolk sac tumour transcriptional similarity per individual, 1000 iterations of random sampling of 6 correlations, matching the size of the PD43299 correlation matrix, generated a distribution of variance against which we could compare the variance of transcriptional similarity in PD43299 (Supplementary Fig. 8).

**Reporting summary**. Further information on research design is available in the Nature Research Reporting Summary linked to this article.

## Data availability

The raw DNA data generated in this study have been deposited in the European Genome-Phenome Archive (EGA) under accession code EGAD00001007038. The raw RNA data generated in this study have been deposited in the EGA under accession code EGAD00001007037. Access to these datasets is restricted due to data privacy laws although access may be granted following an application to the Data Access Committee. The processed data are available in the Article and Supplementary Information. Where the underlying data is not contained within the Supplementary Information, Source data are provided with this paper. Source data for Fig. 3a–c and the read counts generated from the RNA sequencing experiments are deposited on Mendeley and can be accessed here: https://doi.org/10.17632/s8v4t5v9g2.1. Histology data is provided in Supplementary Figs. 1 and 6. Source data are provided with this paper.

## Code availability

The R scripts used to run the bespoke filtering and analyses detailed in this study can be found here: https://github.com/trwo/GCT_diversification.

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

## Acknowledgements

We would like to thank Maxime Tarabichi for reviewing our copy number timing analyses, as well as Peter van Loo for providing valuable advice on our comparisons against TCGA and PCAWG reference data. We thank the CCLG Tissue Bank, the CCLG centres and the ECMC Paediatric Network for the collection and provision of tissue samples (project numbers 2002 BS 03, 2016 BS 05 and 2020 BS 02). The CCLG Tissue Bank is funded by Cancer Research UK and CCLG. Some tissues were acquired from the Human Research Tissue Bank at Cambridge University Hospitals NHS Foundation Trust, whom we must additionally thank for their provision of histological services. The Human Research Tissue Bank is supported by the NIHR Cambridge Biomedical Research Centre. Other research samples were obtained from the Manchester Cancer Research Centre (MCRC) Biobank, UK. The role of the MCRC Biobank is to distribute samples and therefore, cannot endorse studies performed or the interpretation of results. Collection of tissues by R.H. was supported by Cancer Research UK ECMC (C9380/A25138) and the Newcastle Biobank. The CUH adult blood sampling had infrastructure support from the Urological Malignancies Programme which is part of the CRUK Cambridge Centre, funded by a Cancer Research UK Major Centre Award (C9685/A25117), and supported by the NIHR Cambridge Biomedical Research Centre. This research was primarily funded by a Wellcome Trust core grant to the Wellcome Sanger Institute and by personal Fellowships from Wellcome to T.R.W.O. and S.B and from Cancer Research UK to R.R. (C66259/A27114). T.R.W.O. was initially also funded through a personal NIHR Fellowship. A.Y.W. is supported by the NIHR Cambridge Biomedical Research Centre. R.H. is a recipient of a PCF Challenge Research Award (ID #18CHAL11; Heer). The authors also acknowledge grant funding from the St. Baldrick's Foundation (reference 358099). We are thankful for support from the Max Williamson Fund and from Christiane and Alan Hodson, in memory of their daughter Olivia. The funders were not involved in study design, data collection or interpretation, or decision to submit for publication. The views expressed are those of the authors and not necessarily those of the NHS, the NIHR or the Department of Health and Social Care. Lastly, we are grateful to the patients who kindly provided the tissue and blood samples that made this study possible.

## Author contributions

S.B., R.R. and T.R.W.O. designed the experiment. T.R.W.O. and A.C. performed the laser capture microdissection. A.Y.W. and N.C. provided GCT histology expertise. Y.H.

assisted with tissue processing and sectioning, and immunohistochemical staining. T.R.W.O. generated the final lists of substitutions, indels, copy number changes and structural variants with assistance from R.S., T.H.H.C., D.C.W. and M.S. H.J. performed the retrotransposition analysis. T.R.W.O. and R.S. performed SBS signature extraction and analysis with assistance from T.H.H.C., T.B. M.D.C.N. and D.L. Copy number timing analyses were done by T.R.W.O., assisted by S.C.D. I.C. provided expertise on calling chromothripsis-like rearrangements. Phylogenetic analyses were performed by T.R.W.O. with contributions from D.C.W. and N.A.P. L.C. and R.R. designed the low-input RNA sequencing pipeline. L.C. and L.D. constructed the mRNA libraries for sequencing. RNA data processing and analysis was undertaken by T.R.W.O. with guidance from M.D.Y. and R.R. M.J.M., P.J.C. and N.C. contributed to study discussions. R.H. collected study samples. T.J.M. assisted with sample acquisition. S.B., T.R.W.O. and R.R. co-wrote the manuscript. S.B. and R.R. co-directed the study.

## Competing interests

The authors declare no competing interests.
