## [Peer review file · Nature Communications]

REVIEWERS' COMMENTS

Reviewer #1 (Remarks to the Author):

I appreciate the authors thorough consideration of my prior feedback, and the inclusion of supplementary analyses to support the claims presented in this study. I have no further feedback and am supportive of publication.

Reviewer #2 (Remarks to the Author):

The authors have done an excellent job responding to the many concerns that were raised by the reviewers of this manuscript. As noted previously, an immense amount of work has gone into these studies. The data are much better presented in that they emphasize their novel findings more clearly than previously. They also have addressed multiple language issues (overly complex phrases) that made the manuscript difficult to follow in places. However, they still have a few places in which it could be cleaned up, some of which are noted below.

1. Abstract – How does the novel signature lend itself to be a ‘potential biomarker for risk stratification’? The authors are overstating the potential translational relevance of their finding. The authors speculate in the discussion that it might differentiate ‘true’ pediatric from ‘adult’ tumors for therapy. However, interestingly, the only tumor in the age 13-17 age range with this signature also has a KRAS mutation, which would indicate aggressive therapy.

2. Line 55-56. No current data support that there are regional genetic changes associated with histological diversification in NSGCT. In fact, all available data suggest that there are not. Please tone down the statement.

3. Line 103-105. The authors continue to suggest that AKT1 mutations and homozygous deletions of PTPRD are typical drivers of germ cell tumors. Respectfully, they are not. The paper they cite to support the AKT1 mutation reports one mutation in 24 ovarian germ cell tumors, and none have been reported

in TGCT. They did not address the concern raised about the homozygous deletion of PTPRD, which also is extremely uncommon.

4. Extended Fig. 3 – The confusion about the Y-axis was because it was not clear that it was copy number – (minus) ploidy. Perhaps it could be rewritten that way.

Minor Comments

Abstract – What does the sentence ‘We found that the extensive diversification of tissues and genetic subclones were not correlated.’ mean? It is not at all clear, and readers should be able to glean the essence of the manuscript from the abstract.

Page 173 – The authors use complex language in various places to describe straightforward concepts, which makes the manuscript denser to read than it should be. For example, ‘no differences in the genetic similarity’ – could be rephrased as similar genetics between biopsies.

Reviewer #3 (Remarks to the Author):

Oliver and Chappell et al. addressed my previous comments.

I have one remaining minor comment regarding Lines 142 to 144. The current version could be misinterpreted as saying all driver mutations were acquired during subclonal diversification. This sentence should instead be re-written as, “Firstly, truncal mutations included all identified driver events and represented the majority of mutations found in each invasive tumour, suggesting subclonal diversification was a relatively late event.”

**Clonal diversification and histogenesis of malignant germ cell tumours
(previously “Fetal origins of malignant germ cell tumour”)**

We thank the Reviewers for their suggestions which we have addressed as detailed below.

Reviewer 1

No.	Comment	Response
1.0	I appreciate the authors thorough consideration of my prior feedback, and the inclusion of supplementary analyses to support the claims presented in this study. I have no further feedback and am supportive of publication.	We thank the Reviewer for their time and valuable input which helped improve this manuscript.

Reviewer 2

No.	Comment	Response
2.0	The authors have done an excellent job responding to the many concerns that were raised by the reviewers of this manuscript. As noted previously, an immense amount of work has gone into these studies. The data are much better presented in that they emphasize their novel findings more clearly than previously. They also have addressed multiple language issues (overly complex phrases) that made the manuscript difficult to follow in places. However, they still have a few places in which it could be cleaned up, some of which are noted below.	We thank the Reviewer for their suggestions to improve the clarity of our presentation of the work. We have addressed their outstanding comments below.
2.1	Abstract – How does the novel signature lend itself to be a ‘potential biomarker for risk stratification’? The authors are overstating the potential	Whole genome sequencing (WGS) is now commissioned for standard-of-care provision in England for all paediatric malignancies. We anticipate this trend continuing across much of the

	translational relevance of their finding. The authors speculate in the discussion that it might differentiate ‘true’ pediatric from ‘adult’ tumors for therapy. However, interestingly, the only tumor in the age 13-17 age range with this signature also has a KRAS mutation, which would indicate aggressive therapy.	rest of the developed world in the coming years. Mutational signature extraction has become a routine genomic analysis which can be applied to the WGS data. We anticipate the novel signature and other known signatures that define the paediatric cases could be readily detected and form the basis for a study examining their prognostic significance. We hypothesise that tumours in adolescents with a “paediatric” signature may be adequately managed with the paediatric treatment protocol rather than that of an adult. We agree with the author’s sentiment that this supposition is far from proven which is why we are keen to emphasise the uncertainty with “may lend themselves as potential biomarkers”.
2.2	Line 55-56. No current data support that there are regional genetic changes associated with histological diversification in NSGCT. In fact, all available data suggest that there are not. Please tone down the statement.	The relevant sentence has been amended. Changes to manuscript: 1) Lines 56 to 58: “Whilst regional genetic changes appear to underpin histological diversification in some cancers (e.g. the childhood kidney cancer, Wilms tumour), no such evidence exists to support this theory in NSGCTs to date.”.
2.3	Line 103-105. The authors continue to suggest that AKT1 mutations and homozygous deletions of PTPRD are typical drivers of germ cell tumors. Respectfully, they are not. The paper they cite to support the AKT1 mutation reports one mutation in 24 ovarian germ cell tumors, and none have been reported in TGCT. They did not address the concern raised about the homozygous deletion of PTPRD, which also is extremely uncommon.	We have amended the sentence to clearly separate the KRAS and KIT drivers from the other events we describe. According to the cBioPortal, just over 3% of TCGA (postpubertal TGCT) samples have a “deep deletion” of PTPRD. We report 1/14 (7%) postpubertal cases have a driver event consistent with this description. We do not consider this finding out of keeping with the literature. It would also be worth noting that previous studies have been dependent on bulk whole exome sequencing which will have a lower sensitivity for calling copy number changes/structural variants, not least because a subset will have low (<40%) tumour purity which we can avoid in invasive tissues by microdissection. Changes to manuscript: 1) Lines 109 to 111: “We identified typical GCT driver events in our cohort, including KRAS

		substitutions and gains of the KRAS and KIT oncogenes, as well as other drivers such as an AKT1 substitution and a homozygous deletion of the tumour suppressor gene PTPRD .”
2.4	Extended Fig. 3 – The confusion about the Y-axis was because it was not clear that it was copy number – (minus) ploidy. Perhaps it could be rewritten that way.	This has been done.
2.5	Abstract – What does the sentence ‘We found that the extensive diversification of tissues and genetic subclones were not correlated.’ mean? It is not at all clear, and readers should be able to glean the essence of the manuscript from the abstract.	This sentence has been rewritten to offer greater clarity to the reader. Changes to manuscript: 1) Lines 37 to 38: “We find no correlation between genomic and histological heterogeneity.”
2.6	Page 173 – The authors use complex language in various places to describe straightforward concepts, which makes the manuscript denser to read than it should be. For example, ‘no differences in the genetic similarity’ – could be rephrased as similar genetics between biopsies.	We have reworded the relevant sentence to provide an alternative phrase to convey genetic similarity within the text. Changes to manuscript: 1) Lines 179 to 181: “The results suggested that biopsies of the same histology were not necessarily more closely related than two biopsies of different tissues in most cases.”

Reviewer 3

No.	Comment	Response
3.0	Oliver and Chappell et al. addressed my previous comments.	We thank the Reviewer for their feedback and time spent helping to improve this manuscript.
3.1	I have one remaining minor comment regarding Lines 142 to 144. The current version could be misinterpreted as saying all driver mutations were acquired during subclonal diversification. This	We have adopted the proposed rewording. Changes to manuscript:

	sentence should instead be re-written as, “Firstly, truncal mutations included all identified driver events and represented the majority of mutations found in each invasive tumour, suggesting subclonal diversification was a relatively late event.”	1) Lines 149 to 151: “Firstly, truncal mutations included all identified driver events and represented the majority of mutations found in each invasive tumour, suggesting subclonal diversification was a relatively late event.”
--	--	---